⊘ | Open Peer Review | Computational Biology | Research Article

# Genomic re-sequencing reveals mutational divergence across genetically engineered strains of model archaea

Andrew L. Soborowski,[1,2] Rylee K. Hackley,[1,3] Sungmin Hwang,[1] Guangyin Zhou,[4] Keely A. Dulmage,[1,3] Peter Schönheit,[5] Charles Daniels,[6] Alexandre W. Bisson-Filho,[7] Anita Marchfelder,[8] Julie A. Maupin-Furlow,[4] Thorsten Allers,[9] Amy K. Schmid[1,2,3,10]

**ABSTRACT** Archaeal molecular biology has been a topic of intense research in recent decades as their role in global ecosystems, nutrient cycles, and eukaryotic evolution comes to light. The hypersaline-adapted archaeal species *Halobacterium salinarum* and *Haloferax volcanii* serve as important model organisms for understanding archaeal genomics, genetics, and biochemistry, in part because efficient tools enable genetic manipulation. As a result, the number of strains in circulation among the haloarchaeal research community has increased in recent decades. However, the degree of genetic divergence and effects on genetic integrity resulting from the creation and inter-lab transfer of novel lab stock strains remain unclear. To address this, we performed whole-genome re-sequencing on a cross-section of wild-type, parental, and knockout strains in both model species. Integrating these data with existing repositories of re-sequencing data, we identify mutations that have arisen in a collection of 60 strains, sampled from two species across eight different labs. Independent of sequencing, we construct strain lineages, identifying branch points and significant genetic events in strain history. Combining this with our sequencing data, we identify small clusters of mutations that definitively separate lab strains. Additionally, an analysis of gene knockout strains suggests that roughly one in three strains currently in use harbors second-site mutations of potential phenotypic impact. Overall, we find that divergence among lab strains is thus far minimal, though as the archaeal research community continues to grow, careful strain provenance and genomic re-sequencing are required to keep inter-lab divergence to a minimum, prevent the compounding of mutations into fully independent lineages, and maintain the current high degree of reproducible research between lab groups.

**IMPORTANCE** Archaea are a domain of microbial life whose member species play a critical role in the global carbon cycle, climate regulation, the human microbiome, and persistence in extreme habitats. In particular, hypersaline-adapted archaea are important, genetically tractable model organisms for studying archaeal genetics, genomics, and biochemistry. As the archaeal research community grows, keeping track of the genetic integrity of strains of interest is necessary. In particular, routine genetic manipulations and the common practice of sharing strains between labs allow mutations to arise in lab stocks. If these mutations affect cellular processes, they may jeopardize the reproducibility of work between research groups and confound the results of future studies. In this work, we examine DNA sequences from 60 strains across two species of archaea. We identify shared and unique mutations occurring between and within strains. Independently, we trace the lineage of each strain, identifying which genetic manipulations lead to observed off-target mutations. While overall divergence across labs is minimal so far, our work highlights the need for labs to continue proper strain husbandry.

**Peer Reviewers** Kylie Allen, Virginia Polytechnic Institute and State University, Blacksburg, Virginia, USA; Houda Baati, University of Sfax, Sfax, Tunisia

Address correspondence to Amy K. Schmid, amy.schmid@duke.edu.

The authors declare no conflict of interest.

See the funding table on p. 14.

**KEYWORDS**    archaea, genomics, mutational analysis, whole genome re-sequencing

In recent decades, hypersaline-adapted archaea, hereafter referred to as haloarchaea, have emerged as leading model organisms in the study of archaeal genetics, genomics, and biochemistry (1–8). Unique among archaeal lineages, this class has produced multiple genetically tractable organisms, facilitating both single-organism studies and class-wide comparisons. Haloarchaeal genomes, which tend to be organized similarly to bacteria, are characterized by genes co-regulated in operons and relatively short non-coding sequences. Considerable research has already been performed in two of these organisms, *Halobacterium salinarum* (*Hbt.*) and *Haloferax volcanii* (*Hfx.*), for which complete genome sequences (9, 10), counterselection-based gene knockout systems (11, 12), additional selectable markers (13), and overexpression plasmids (14), among other tools, have been developed (15). *Hbt.* has been studied using related laboratory strains NRC-1 and R1, though for this paper, we focus exclusively on strain NRC-1 (16, 17). Both species contain a single, circular chromosome containing the majority of their genes, in addition to a small number of additional replicons, pNRC100 and pNRC200 in *Hbt.* and pHV1, pHV2, pHV3, and pHV4 in *Hfx.* (9, 10). These replicons, with the exception of the pHV2 plasmid, are referred to as mega-plasmids as they are large, contain essential genes, and contain functional origins of replication (18, 19). However, the mega-plasmids are enriched for repetitive sequence and duplicated genes, complicating genomic assembly.

Gene knockouts have become commonplace in the study of *Hbt.* and *Hfx.*, owing to their relative simplicity given the power of existing toolkits. Current methodology in both species uses a selection/counterselection ("pop-in/pop-out") approach (12). The system exploits the uracil biosynthesis pathway by generating unmarked mutations in two steps of selection, similar to that used commonly in yeast genetics. This system facilitates a relatively easy process to generate gene knockouts using a uracil auxotroph strain as the parent from which subsequent mutant strains are derived. These strains are commonly used in both species, and we will refer to them as Δ*ura3* in *Hbt.* and Δ*pyrE2* in *Hfx.* (11, 12).

At the conclusion of the pop-in/pop-out procedure, it is crucial to verify the successful deletion of the gene because of the following: (i) the procedure is equally likely to result in the parental and the desired knockout genotypes. If the desired knockout is deleterious, the parental strain revertant is even more likely; (ii) secondary mutations arising during selection can alter the phenotype observed in any downstream experiments, confounding results; (iii) haloarchaeal genomes are polyploid, harboring approximately 20 genome copies in a single cell during exponential phase (20, 21). Polyploidy enables low-level wild-type gene copies to persist during and after selection if the gene is essential. Genotypic verification is often performed using end-point PCR to amplify across the genomic region targeted for deletion, or a Southern blot to probe sequences flanking the deleted region. However, recent research in our lab and others has shown that whole-genome DNA re-sequencing (WGS) provides the sensitivity required to detect second-site mutations or low-level wild-type copies of the gene that remain in the polyploid genome below the level of detection by PCR (22–29). Follow-up experiments to determine the phenotypic consequences of any detected second-site mutations are therefore important. For example, using WGS, we have identified multiple second-site mutations, some with functional consequences (e.g., second-site mutations in the novel TF-coding gene *tbsP* suppresses lethality of *trmB* deletion under gluconeogenic conditions) (22–24). In contrast, other second-site mutations did not affect the primary phenotype of interest (22). Nonetheless, low levels of a wild-type copy of the gene of interest and/or second-site mutations undetectable with PCR or Southern blotting remain a primary concern in genetic manipulation of polyploid organisms. In a post-genomics world, WGS has become increasingly affordable, and thus increasingly suitable, for this task with the recent NovaSeq X promising a $200 human genome, approximately $2.00 per Gb of short reads. Long-read sequencing is also an option,

trading cost and marginal accuracy for the ability to detect large-scale rearrangements and easily assemble across repetitive regions.

WGS is also an important method for periodic verification of the genetic integrity of the wild-type and parental strains. Although genetic analysis is typically done in relation to a published reference sequence, there is always a possibility of novel mutations arising in an individual lab's stock due to either random drift or adaptation to the lab environment. Owing to the collaborative nature of halophile research, it is common for labs to transfer strains to other labs. This process involves inoculating either a glycerol stock or agar stab in the lab of origin, shipping the sample to the recipient lab, culturing the received cells, then generating a long-term glycerol freezer stock. Like genetic manipulations, this process produces a genetic bottleneck, allowing mutations to fix in the final population depending on the colony selected. As stock strains are continually passaged and transferred to new labs, the possibility increases for functional mutations to arise and propagate, confounding comparisons both to the static reference and comparisons made between labs. This concern is already well realized among labs studying common model organisms (30, 31), but this problem has yet to be addressed among haloarchaeal species. WGS via short-read next-generation sequencing therefore presents a relatively inexpensive, labor lite, and highly sensitive approach to address these concerns that are not detectable with PCR-based genetic verification of strains.

In this study, we report WGS data for 60 strains (51 of which have not previously been published) across both model haloarchaeal species with at least 29-fold coverage. We focused primarily on gene deletion strains without prior sequencing data to verify genetic backgrounds for use in further experiments, both in our lab and in the greater community. Additionally, we sequenced wild-type and uracil auxotroph parental strains across both species and sourced from many lab groups. This enabled the identification of mutations that have arisen at each step between the published reference strain and the final knockout, as well as any divergence that may have arisen between parental lab strains. With these data, we demonstrate that overall divergence between lab strains has thus far been minimal, though both lab-to-lab transfers and purposeful genetic manipulations generate mutations fixed between parental strains. Additionally, we identify cohorts of mutations fixed among each sampled strain in contrast to the published reference, as well as mutations fixed in distinct lineages. A survey across knockout strains in both species reveals a number of second-site mutations of potential phenotypic impact, highlighting the need for WGS verification when performing genetic manipulations.

## RESULTS

### Lineage tracing points to recent shared ancestry of most strains

To understand the genome-wide mutational landscape across laboratory domesticated haloarchaea, we performed short-read next-generation whole genome DNA re-sequencing (WGS) on a collection of 60 parental and gene knockout strains of *Hbt*. and *Hfx*., compiling a data set of unique mutations as identified by the microbial mutation caller breseq (32) (see Materials and Methods). Since many of the strains in this collection were sourced from different labs, we first traced the lineage of each strain by combining a thorough literature review and personal correspondence. We traced commonly used parental strains back to the commercially available wild-type strain for both *Hbt*. (Fig. 1A) and *Hfx*. (Fig. 1B), identifying both genetic manipulations performed to create new parental strains and direct transfers of parental strains between labs. To distinguish between identical background strains sourced from different labs, here we label each parental strain with the lab's initials. These labels correspond to the lab in which the strain was used in subsequent work, though we note that every strain but one was subsequently passaged and transferred to the Schmid lab for sequencing (Table S1).

The *Hbt*. wild-type strain used in this study is NRC-1, first sequenced in 2000 and commercially available via ATCC (700922) (9). Δ*ura3* is a uracil auxotroph and the sole *Hbt*. parental strain included in this study. Generated from NRC-1, Δ*ura* was also

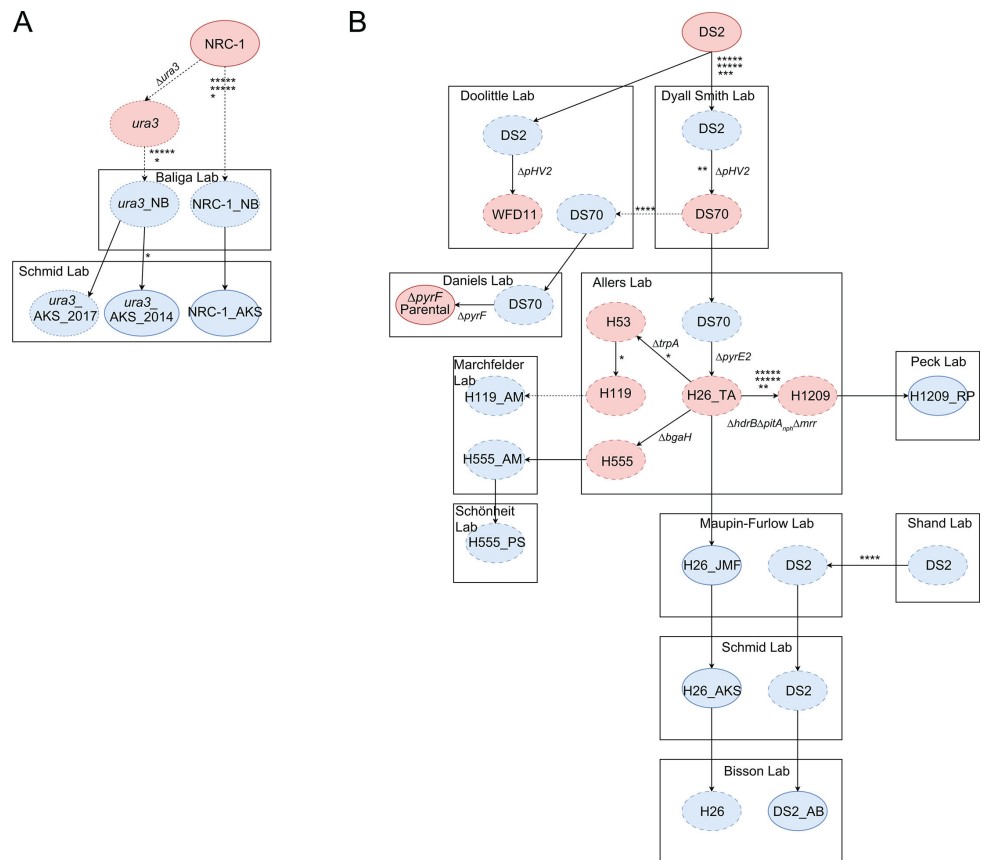

**FIG 1** Lineage map of wild-type and parental strains. (A) *Hbt.* and (B) *Hfx.* diagrams of the lineage of sequenced and related parental and wild-type strains. Circles with solid outlines correspond to sequenced strains, whereas circles with dashed outlines correspond to known intermediate strains not sequenced here. Rectangles demarcate strains constructed within each lab. Full arrows correspond to known genetic manipulations (within rectangles) or transfers between labs (rectangle to rectangle). Dashed arrows refer to implied events with unknown intermediate steps. Sequenced parental or wild-type strains are assigned a unique postscript corresponding to their lab. Red circles correspond to the first instance of a given lineage, whereas blue circles depict strains transferred between labs. Asterisks denote individual mutations and are placed at the earliest transition they may have occurred. Unique identifiers for Δ*ura3*, Δ*pyrE2*, and Δ*pyrF* are given in Table S1.

developed in the year 2000 to enable the construction of unmarked deletion mutations (11). Descendants of both strains were transferred to the Baliga lab, which we identify as NRC-1_NB and Δ*ura3*_NB. Δ*ura3*_NB was used as the parental strain to construct 13 knockout strains included in this study. Descendants of Δ*ura3*_NB were transferred to the Schmid lab in 2009 and later sequenced in 2014 (Δ*ura3*_AKS_2014, HS149) and 2017 (Δ*ura3*_AKS_2017, HS148). NRC-1_AKS, a descendant of the wild type, was also sequenced in 2014. Δ*ura3*_AKS_2014 was used to construct five knockout strains included in this study, and Δ*ura3*_AKS_2017 was used to construct one knockout strain included in this study.

We identified 11 mutations common across each *Hbt.* strain that differed from the published reference sequence (9), suggesting that these mutations arose: (i) before Δ*ura3* was generated from NRC-1, (ii) in the specific NRC-1 isolate used to produce the reference genome (Fig. 1A), or (iii) from errors in the reference genome. Additionally, we identified six mutations common to every strain except NRC-1_AKS. As the most recent common ancestor of these strains is Δ*ura3*_NB, and the mutations are undetected in the wild-type strain, it is most likely these arose alongside the knockout of *ura3* or during a previous transfer of the strain. An additional mutation was uniquely identified in all descendants of Δ*ura3*_AKS_2014, but absent in every other sequenced *Hbt.* strain. The distribution of this mutation among the sequenced strains suggests it is fixed in the

long-term freezer stock of Δ*ura3*_AKS_2014 and either absent or present in a minority of cells in the Δ*ura3*_NB. This is because the mutation was detected in every descendent of Δ*ura3*_AKS_2014, but in no other descendants of Δ*ura3*_NB. The most likely explanation is that the mutation arose, or at least swept the population, during the transfer of Δ*ura3*_AKS_2014 to the Schmid lab.

In the *Hfx*. lineage, the wild-type strain used in this study is DS2, sequenced in 2010 and available commercially via ATCC (29605) (10). To enable genetic manipulation in *Hfx*., DS2 was cured of the pHV2 plasmid to give rise to the DS70 parental strain created in the Dyall Smith lab (33). Subsequently, due to parallel efforts to improve DS70 for the construction of genetic knockout strains, the lineage of DS70 split, resulting in the Daniels lab Δ*pyrF* strain (Δ*pyrF*_CD) and the Allers lab Δ*pyrE2* (H26_TA) strain. H26_TA serves as the ancestor of all knockout strains included in this study. Labs included in the current analysis either received their copy of H26 directly or secondarily (via an intermediate lab) from the Allers lab. Additionally, the Allers lab developed further descendants of H26_TA to facilitate genetic work, including H119, H555, and H1209 (13, 14). We refer to H26, H119, H555, H1209, and Δ*pyrF*_CD as parental strains for the current analysis.

For *Hfx*., all knockout strains sequenced for this study are derived from the following five parental strains: the sequenced H26_AKS, H26_JMF, and H1209_RP, as well as the unsequenced H119_AM and H555_PS (Fig. 1B; Table S1). Each of these parental strains shares the Allers lab H26 strain as a common ancestor. In addition, wild-type DS2_AB and the Δ*pyrF*_CD parental strains were sequenced, although no knockouts derived from these are included here. DS2_AB is known to have been sequentially transferred between at least four labs, with the first known transfer occurring in 1999 from the Shand lab to the Maupin–Furlow lab (34).

Across these strains, we identified 13 common mutations. Like in *Hbt*., these findings suggest that these mutations arose in the common ancestor of the DS2 lineage, in the specific DS2 strain used to construct the reference genome, or they represent errors in the assembly (Fig. 1B). Additionally, we identified two mutations in every strain except DS2_AB, suggesting they arose after the last common ancestor but before the lineage split between H26_TA and Δ*pyrF*_CD. We identified four mutations unique to the Δ*pyrF*_CD lineage but none unique to the H26_TA lineage. Among the five parental strains from which the knockout strains were directly created, we identified no mutations in the H26_JMF, H26_AKS, or H555_PS lineages, though H26_AKS does carry two mutations not detected in any descendent strains, likely fixed only in the specific colony used for re-sequencing. The H119_AM lineage carries two mutations and the H1209_RP lineage carries 12. Taken together, these results demonstrate that, for both species, mutations arise during the process of routine laboratory cultivation, generation of deletion strains, and inter-lab transfers.

## Class I mutations are shared across all strains tested

To further investigate mutations arising during inter-lab strain passages and within-lab genetic manipulation, we tabulated unique mutations (see Materials and Methods for how mutations were identified) appearing in each strain of the species *Hbt*. (Fig. 2A) and *Hfx*. (Fig. 2B). Across both species, mutations were separated into three classes. Class I was mutations detected in every strain of a given species, i.e., different from the published reference sequence. Class II was mutations unique to a parental strain and subsequently derived knockout strains. Class III was mutations unique to a specific knockout strain acquired during the process of strain construction. Additionally, mutations were further delineated between mobilome (mobile genetic elements, see Materials and Methods for definition) and non-mobilome-related changes (Fig. 3A and B). Detailed annotations for each reported mutation are given in Table S2 for *Hbt*. and Table S3 for *Hfx*.

For *Hbt*., 11 total mutations were detected in the first class of mutations (shared across strains, Fig. 1A, 2A, and 3A) (9). Of these, six are located within mobile elements

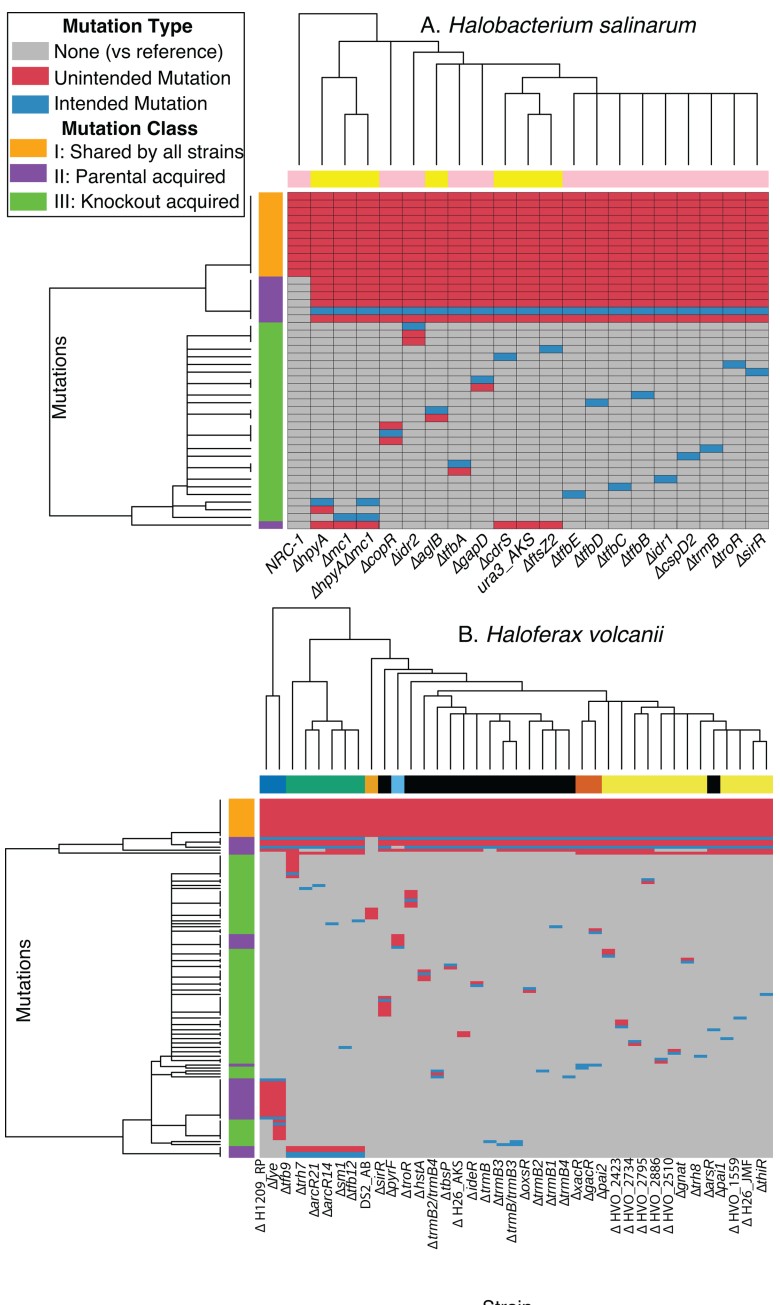

**FIG 2** Heatmap of presence of mutations in (A) *Hbt.* and (B) *Hfx.* Across both panels A and B, gray entries correspond to mutations not present in the strain, blue entries correspond to experimenter-intended mutations that are present in the strain, and red entries correspond to mutations that are not intended to be made in the strain. Colors on the vertical bars represent the classes of mutations: orange represents Class I mutations shared by all strains (that differ from the published type strain (10); purple indicates Class II, acquired in parental strain(s); green indicates, Class III mutations, acquired only in the knockout strains. Horizontal bars represent the lab of origin. For *Hbt.*: yellow, Schmid lab; pink, Baliga. For *Hfx.*: dark blue, Peck; green, Marchfelder; light orange, Bisson; black, Schmid; light blue, Daniels; dark orange, Schönheit, yellow, Maupin–Furlow. Clustering was performed on both strains and mutations with mutation clustering only on the presence or absence of the gene.

(Fig. 3A). Four of these mobilome mutations cluster in the same gene, VNG_RS00125, predicted to encode an IS5 family transpose. The gene is located in a highly repetitive region of the genome; however, the mutations occur in sequence unique to the gene.

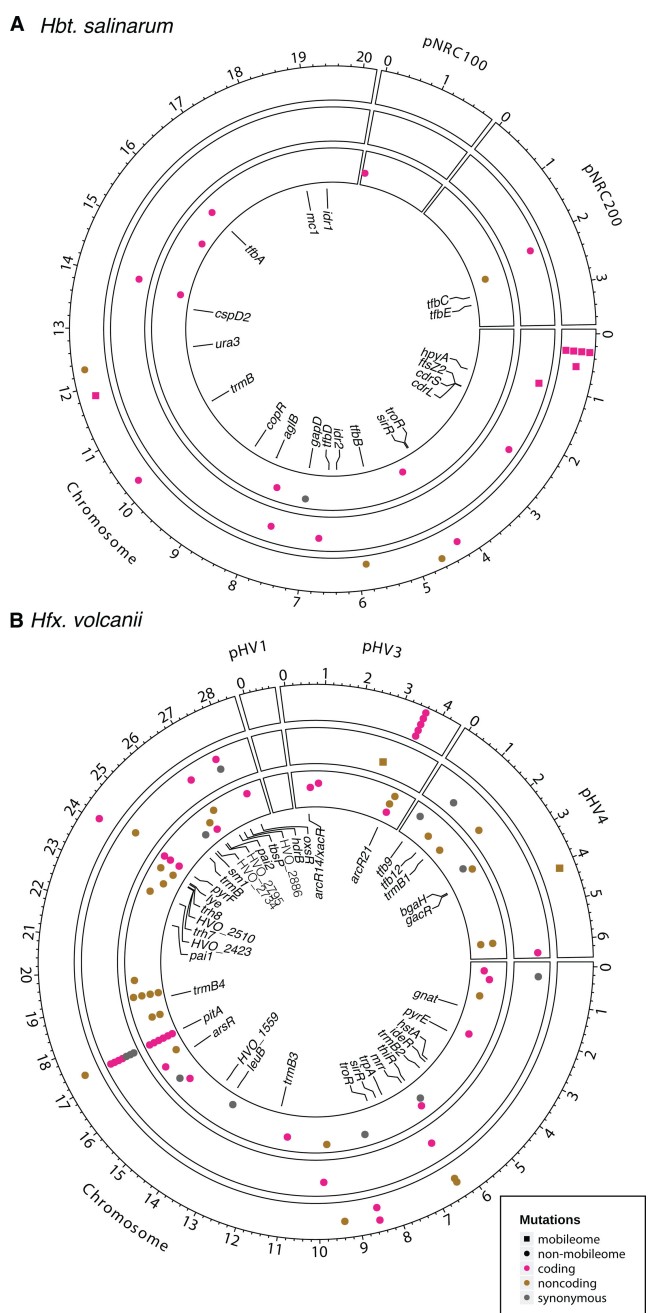

**FIG 3** Genome plot of detected mutations in (A) *Hbt*. and (B) *Hfx*. Genomic distance is measured in Mbp. Each point corresponds to a unique mutation. Sequence features are as follows: black, non-mobilome; red, coding region; green, non-coding regions. Mutation types: grey, synonymous mutations; pink and tan, non-synonymous mutations, grouped according to coding (pink) and non-coding (tan) mutations; see also pictorial legend. Mutations in the mobilome are represented as squares. Points are localized to the chromosomal coordinate at the start of the corresponding mutation. Concentric rings correspond to mutation class, with the outer corresponding to class I, the middle class II, and the inner class III. Gene names correspond to intended knockouts.

Two mutations resulted in the deletion of VNG_0059H and VNG_1587H, annotated as IS4-like element ISH8B and IS5-like element ISH11, respectively, for which exist multiple identical and nearly identical duplicates in the genome. A closer examination revealed read troughs in the unique sequence directly upstream and downstream of each gene, consistent with a transposition event. Of the remaining five mutations, one may be of

functional significance: a single-nucleotide polymorphism (SNP) was detected in the coding region of VNG_1374G, which changes a serine to an arginine codon (Table S2; Fig. 3A). This gene is annotated as a PAS domain S-box protein, which may be involved as a sensor module or transcriptional activator. A single nucleotide insertion was also detected in the intergenic region between VNG_1651H and VNG_1653H. The remaining three mutations correspond to the loss of a single base in a poly-C or poly-G tract region in a hypothetical protein and two pseudogenes. These are likely the result of sequencing errors (either here or in the originally published type strain genome sequence) given the technical challenges of sequencing homopolymer tracts (35). Taken together, these results suggest the published NRC-1 reference sequence (9) still accurately represents strains currently in use. Moreover, many of the observed differences occur either in highly repetitive sequences or involve polynucleotide runs, which may be indicative of sequencing error or transposon movement with little impact on phenotype.

For *Hfx.*, 13 mutations belong to the first class (shared across all strains, Fig. 1B and 2B). One predicted mutation, a SNP in HVO_2547 (changes L→P in a 50S ribosomal protein) was previously characterized and determined to be an error in the reference genome (36). Three mutations were detected as single base indels either in intergenic regions (between HVO_1027 and HVO_1028, and between HVO_3045 and HVO_1878), distant from transcription start sites, or within mobile elements (HVO_A0408, encoding an ISH3 family transposase; Fig. 3B and Table S3). These are unlikely to have functional consequences. The remaining nine of these 13 mutations represented single base indels or SNPs that occur within or upstream of genes critical for metabolism and stress response: five immediately upstream or within the coding region of HVO_B0311 (XdhC/CoxL family protein), two 64 bp apart within HVO_0712 (shikimate dehydrogenase), and two 9 bp apart within the coding sequence of HVO_0938 (universal stress protein). Therefore, unlike in *Hbt.*, the majority of these *Hfx.* class I mutations could have functional consequences in gene expression and/or protein function.

## Class II mutations were acquired in the parent strains during transfers or genetic manipulation

In *Hbt.*, six class II mutations are shared in the strains descended from the Baliga lab Δ*ura3* strain but undetected in the ancestral NRC-1 strain (Fig. 2A) (9). One corresponds to the expected deletion of the *ura3* gene. One of the remaining five unexpected mutations involves the deletion of an ISH3-like element ISH27-2 family transposase (VNG_RS00465), which may have also deleted a small number of bases in a hypothetical protein 20 bases upstream (Fig. 3A). Three mutations are SNPs inducing amino acid changes in three distinct genes: *trpB*, *gyrB*, and VNG_6283H, a hypothetical protein located on the pNRC200 plasmid. *trpB* codes for subunit beta of the tryptophan synthase protein complex, while *gyrB* is predicted to code for subunit beta of a DNA topoisomerase involved in DNA supercoiling (37, 38). Finally, a 7-bp deletion was detected in VNG_1896C (predicted ATP-binding protein) exclusively in strains Δ*ura3*_AKS_2014 and its descendants (Δ*hpyA*, Δ*mc1*, Δ*hpyA*Δ*mc1*, Δ*cdrS*, Δ*ftsZ2*). This suggests that the VNG_1869C mutation arose during the transfer of the strain from the Baliga lab to the Schmid lab. Taken together, these six mutations in the second class suggest the potential for phenotypic differences between derivative strains and their corresponding parent. However, to our knowledge, no phenotypic differences between these parent strains aside from the intended uracil auxotrophy of the Δ*ura3* strain have yet been observed or reported.

In *Hfx.*, 30 total class II mutations were detected. One intergenic mutation was detected exclusively in reads over 150 bp and one pseudogene mutation was detected randomly in a highly repetitive region; both were excluded from further analysis. Four were shared between each strain derived from H26_TA (Fig. 2B): (i) expected deletion of the pHV2 plasmid and *pyrE2* (Fig. 3B), and (ii) one SNP in each of HVO_0032 and HVO_1080 (Fig. 3B). In contrast, five mutations were specific to the Daniels lab Δ*pyrF* strain, necessarily arising either during transfer to the Daniels lab or during the knockout

of the *pyrF* gene within the Daniels lab. These mutations include the expected deletion of *pyrF*, a SNP in *ldpA* (dihydrolipoyl dehydrogenase), and three mutations, including a frameshift in HVO_2576, a J-domain-containing protein potentially involved in protein assembly and translocation. Additionally, this lineage also lacks pHV2 and carries the HVO_0032 and HVO_1080 mutations, suggesting they originated before the H26 and Δ*pyrF* lineages split.

Among the parental lineages descended from H26_TA (H1209_RP, H26_JMF, H26_AKS, H555_PS, H119_AM), we detect 19 additional, lineage-specific mutations (Fig. 1B, 2B and 3B). Five of these are expected, corresponding to the gene knockouts required to make the H119, H555, and H1209 strains. An additional four mutations were specific to the H1209_RP lineage, which may correspond to difficulty for breseq to properly recognize the replaced copy of *pitA* (14). Of the remaining 10 mutations, three correspond to synonymous changes and one to an intergenic SNP (Fig. 3B). Four nonsynonymous SNPs were detected in the H1209_RP lineage. Two nonsynonymous SNPs were detected in the H119_AM lineage, occurring in metallopeptidase HVO_A0634 and calcium/sodium antiporter HVO_0714. The mutation in HVO_A0634 was previously identified in H53, an intermediate between H119 and H26 (36). No additional mutations were detected in the H26_JMF, H26_AKS, or H555_PS lineages. Taken together, these mutations in both species highlight a growing, though as yet small, divergence between the parental strains at a genetic level with unknown phenotypic consequences as subsequent genetic manipulations are performed in their descendants.

## Class III mutations, acquired during construction of deletion strains, may lead to phenotypic consequences

In *Hbt.*, a total of 19 knockout strains were sequenced, including 18 single knockouts and one double knockout. We identified a total of eight unique, unexpected mutations across these strains. Each strain averaged 0.42 unexpected second site mutations with no strain having more than two (Fig. 2A). Of these, six mutations alter a coding sequence, one is synonymous, and one occurs in an intergenic region (Fig. 3A). Of the six coding mutations, three consist of mutations in hypothetical genes: a SNP in VNG_2183H, a 6-bp polynucleotide duplication in VNG_1026H, and an 87-bp deletion in VNG_0597H. The remaining coding mutations consist of an ATG–ATA start codon SNP in *nadA*, a SNP in transcriptional regulator *lrp*, and the complete elimination of the pNRC100 plasmid. Detailed descriptions of each mutation are given in Table S2.

In *Hfx.*, a total of 34 knockout strains were sequenced, 32 single knockouts and two double knockouts. We identified a total of 41 unique, unexpected mutations across these strains (Fig. 2B). Six of these correspond to mutations in HVO_1871, which are likely an artifact used to create the parent strain, H1209_RP (14). Of the remaining 35 mutations in 33 strains, each strain has at most seven mutations, though, on average, each strain has only 0.95. Of these 35 mutations, eight alter coding sequences, while six are synonymous, and 21 occur in noncoding sequence (Fig. 3B). There are only two strains in which a mutation occurs within 50,000 bp of the intended knockout site. Each coding mutation affects only a single gene and is either a SNP or a short (9–41 bp) deletion. Detailed descriptions of each mutation are given in Table S3.

Overall, unintended mutations in *Hbt.* knockout strains appeared in only six of the 19 (32%) strains sampled, although they are likely to lead to phenotypic consequences due to their prevalence in coding sequence. Mutations appear more frequently in *Hfx.*, appearing in 16 out of 33 (48%) strains. While overall mutation density is also higher in *Hfx.* at 0.95 mutations per strain versus only 0.42 mutations per strain in *Hbt.*, *Hfx.* tends to accumulate more noncoding and synonymous mutations. Coding mutation density is similar between the species, with *Hfx.* accumulating 0.24 coding mutations per strain and *Hbt.* accumulating 0.32. These coding mutations are the most likely to cause phenotypic impact, and they affect a broad range of genes, including enzymes, transporters, transcription factors, and genes of unknown function.

## DISCUSSION

This study provides a significant expansion in the available number of short-read whole-genome DNA sequencing data sets available in the Haloarchaea, adding 51 novel strain sequences across *Hbt.* and *Hfx*. Analyzing these sequences alongside previously described sequences using the mutation caller breseq, we identify a population of mutations present in each of the species. By combining identified mutations with cataloged lineage information, we pinpoint the time and intermediate strain in which each mutation likely arose. This allows the classification of mutations according to their prevalence across all sequenced strains, enabling both the analysis of extant mutations and their potential phenotypic impacts, as well as prediction of the type and frequency of mutations likely to arise in consort with common laboratory operations.

Global mutations occurring in every sequenced strain were detected 11 times in *Hbt.* and 13 times in *Hfx*. While it is difficult to conclude in which lineage these arose, three scenarios are possible: (i) mutations arose in the last common ancestor of every sequenced strain, (ii) mutations arose independently in the single isolate used to generate the reference sequence, or (iii) sequencing errors. In the case of both species, these mutations tend toward repetitive elements, noncoding changes, and multiple hits to the same gene. Parental mutations, occurring in distinct lineages among the sequenced strains, were detected in the uracil auxotroph lineages in both species, Δ*ura3* in *Hbt.* and H26 in *Hfx.*, as well as many of the branching *Hfx.* parent strains derived from H26. Many of these changes tended to be coding, including every parental mutation in *Hbt*. Knockout mutations, occurring uniquely in single strains or derived double knockout strains, were detected in 32% of *Hbt.* strains and 48% of *Hfx.* genes. Both strains accumulated a similar number of coding mutations per knockout, though *Hfx.* strains were more likely to accumulate other synonymous or non-coding changes. This analysis highlights the importance of maintaining a permanent frozen stock within each lab to avoid accumulating mutations during routine experiments and inter-lab transfers.

In categorizing the mutations into three classes, we identify three experimental activities in which mutations commonly arise: parental strain generation, inter-lab transfers, and knockout strain generation. Background mutations are expected by chance. For example, the background mutation rate for *Hfx.* is estimated at $3.15 \times 10^{-10}$ per site per generation (39). However, these three lab experimental activities involve selection via population bottlenecks during colony selection steps. Bottlenecking is then compounded with selective pressure during strain generation, allowing both random and selected mutations to fix after the bottleneck step. As we only sequence a small number of strains compared to the large number of intermediates that may have existed, it is often difficult to conclude when a mutation became fixed. For example, in *Hbt.*, a mutation present in Δ*ura3*_AKS_2014 and all its descendants but not Δ*ura3*_NB, provides evidence that lab transfers can result in fixed mutations. However, the lack of mutational enrichment in strains transferred to the Schmid lab for sequencing, and thus undergoing an additional lab transfer step, vs native Schmid lab strains that have not suggests that many of the mutations we observed likely arose during strain generation steps and not as a result of the transfer process. Regardless of mutational source, our analysis suggests that, while overall divergence between strains remains low, phenotypically impactful mutations can and do arise. This highlights the importance of WGS genotyping coupled with complementation experiments to ensure validity and reproducibility of the genotype–phenotype relationship in both intentionally generated mutants and primary lab stock.

Our use of short-read sequencing data in combination with breseq imposes two major limitations in this analysis. Breseq was chosen as the mutation caller in the study as it is the primary tool used in our lab and others for analyzing genomic DNA for mutations. Breseq has successfully called second-site mutations in recent genomic analyses across the tree of life (22–24, 40–42). However, the first limitation imposed by our methodology is that the detection of large-scale rearrangements and mutations in highly repetitive regions is problematic, as it is difficult to place rearrangement

junctions with short reads and uniquely map short reads in highly repeated sequences. Previous work in *Hbt*. has already shown rearrangements among repeat sequence families to be fairly common, especially on the megaplasmids: on the order of $10^{-3}$ potential arrangements occur per repetitive sequence family per generation (15). Recent work in *Hfx*. has also demonstrated a high propensity for genomic rearrangement, including division of the main chromosome into independent, self-sufficient replicons in the course of performing serial gene deletions (43). While breseq attempts to predict genomic rearrangements and other large-scale events, predictions are of lesser quality and uncertain at boundaries. Future studies that incorporate long-read sequencing data, either exclusively or in tandem with short-read data, would be particularly fruitful in this regard, especially as the accuracy of these methods is rapidly improving (44).

The second limitation imposed by our methodology is a lack of identification of mutations that are transient in the population during transfer and strain construction. In theory, the population of cells should be clonal after each genetic bottleneck event, resulting in an entire population of genetically identical cells, such that sampling the DNA sequence of any sub-sample is equivalent to any other sub-sample. However, the cells are not static after the bottleneck event, where we expect some variation to arise, but not become fixed, in the final population. Additionally, both species are known to be highly polyploid, carrying up to 20 copies of the chromosome (45). Heterozygous states transiently exist in the absence of selection but can endure in the presence of selection (46). Problematically, from strictly looking at aggregated DNA sequences, it is not possible to differentiate inter-chromosomal differences versus inter-cellular differences, preventing the identification of heterozygous mutations and their possible phenotypic effects. Tracking such transient low-level variants over time is therefore an interesting avenue to pursue in future research.

Supported by independent lineage tracing, we demonstrate that functional divergence between lab strains has, thus far, remained minimal, though mutations of potential phenotypic impact have begun to arise as strains are continually transferred and manipulated by different lab groups. Additionally, we demonstrate a consistent accumulation of coding mutations in knockout strains in both species. Taken together, our results highlight the need for vigilance when working with haloarchaeal strains. We recommend minimizing transfer events that lead to strain divergence by obtaining strains from a common, ancestral source. When creating a new lab stock after a transfer or genetic manipulation, multiple candidate colonies should be selected. Strain integrity of each candidate should be verified using whole-genome re-sequencing and knockouts double-checked via complementation assay such that the candidate with the least divergence is chosen. To avoid accumulating mutations within a lab, we recommend housing all strains in glycerol in permanent storage at −80°C (see Halo handbook at https://haloarchaea.com/halohandbook/). Strains should be refreshed from frozen stock prior to each and every experiment instead of being serially passaged on plates. As the study of haloarchaea continues to expand in the coming decades as a core model system, these measures will facilitate reproducible science by reducing unintended strain divergence in both knockout strains and core lab stocks.

## MATERIALS AND METHODS

### Strain construction protocols

To construct knockout (KO) mutant strains of *Hfx. volcanii*, the double crossover selection method ("pop-in/pop-out") was applied (12, 13, 46). First, knockout (KO) plasmids were generated. Briefly, approximately 500 bp of the flanking regions of the sequence to be deleted were cloned into pTA131. For example, for Schmid lab vectors, pre-KO primers were used to generate PCR amplicons harboring the genomic regions flanking the target gene of interest (approximately 500 bp each, see primers for all labs and strains in Table S1). These amplicons were then digested by appropriate restriction enzymes and inserted into the multiple-cloning sites of pTA131, resulting in "pre-KO" plasmids. For

deletion strains generated in the Marchfelder laboratory (Δ*trh7*, Δ*tfb9*, Δ*tfb12*, Δ*arcR14*, and Δ*arcR21*), the genomic region flanking each target gene (approximately 500 bp) were amplified as a single fragment by overlap PCR, resulting in a single DNA fragment cloned into pTA131. These plasmids were utilized as templates to generate KO plasmids via reverse PCR with KO primers. For all labs, KO plasmids were transformed into *E. coli dam⁻* strain GM2163 prior to transformation of *Hfx. volcanii* Δ*pyrE2* strain background (see Table S1 for the specific strain number used by each lab). As described in the Halohandbook (https://haloarchaea.com/halohandbook/), the polyethylene glycol (PEG) method was used for transformation followed by selection of pop-in transformants on Hv-CA plates without uracil. Pop-out transformants were subsequently selected on YPC plates supplemented with 50 µg/mL 5-fluoroorotic acid (5-FOA). Clones were screened by PCR primers listed in Table S1, then subjected to whole-genome re-sequencing as detailed below. The Daniels lab *Hfx. volcanii pyrF* gene was deleted using the "pop-in/pop-out" strategy and checked using primers listed in Table S1. For *Hbt. salinarum* KO strains, the same protocol was followed except that Δ*ura3* was the parent background (11), pNBK07 the cloning vector (47), and complete medium (CM) was used for routine growth (see reference 28 for recipe).

## Sources for strains reported in this study

For *Hbt.*, strains were sourced from the Schmid and Baliga labs: a wild-type and 13 knockout strains from the Baliga lab (48–53), and a parental (11) with six knockout strains from the Schmid lab (23, 26, 28, 54, 55).

For *Hfx.*, strains are sourced from seven labs: a wild type from the Bisson lab, a parental strain from the Daniels lab, six knockout strains from the Marchfelder lab (56), a parental strain and 11 knockout strains from the Maupin–Furlow lab (12, 57), two knockout strains from the Schönheit lab (58, 59), a knockout and corresponding parental strain from the Peck lab (14, 60), and 14 knockout strains with the corresponding parent strain from the Schmid lab (12, 22, 24, 27, 61). Note that parent strains for *Hfx.* originated from the Mevarech (12) and Allers labs (14), but were not resequenced here (see Results).

## Cultivation of strains, gDNA extraction, and sequencing

*Hfx. volcanii* strains (see Table S1 for complete strain list and additional sequencing details) were routinely grown by streaking from frozen glycerol stock onto solid YPC18% medium supplemented with 50 µg of uracil and grown at 42°C (see reference 14 for media formulation), except for *trmB* knockout strains, to which 1% glucose was added (49). Genomic DNA (gDNA) was extracted from mid- to late-exponential phase YPC18% liquid cultures using 25:24:1 phenol:chloroform:isoamyl alcohol and ethanol precipitation as described previously (26). *Hbt.* strains were treated similarly except that the medium used was CM medium containing 50 µg of uracil (see reference 23 for media formulation). For both species, gDNA was subject to MiSeq or Illumina whole-genome sequencing by the Duke Sequencing and Genomic Technologies Core Facility (see Table S1 for further information about sequencing platform). For reads sequenced on a MiSeq v3 platform to produce 300-bp PE reads, DNA was first sheared to an average size of 500 bp using the Covaris S2 sonicator, and sequencing libraries were built using the KAPA HyperPrep kit (Roche Material Number: 07962363001). Raw sequence files for nine previously sequenced strains were obtained from the following sources (22, 25, 27, 61) and analyzed identically to the sequence files generated in this study.

## Identifying mutations

Before sequence alignment and variant calling, adapter sequence was trimmed and low-quality reads removed using Trim Galore (62). Sequences for each species were aligned to their respective reference genome (*Hbt.* at https://www.ncbi.nlm.nih.gov/datasets/genome/GCF_000006805.1/ accessed on 19 December 2022 and *Hfx.* at https://www.ncbi.nlm.nih.gov/datasets/genome/GCF_000025685.1/ accessed on 12 September 2022) using Bowtie2 (63) with mutations called by the microbial variant caller breseq

using default parameters and run in clonal mode (32). Mutational calls were tabulated using the supplemental tool GDtools (32). Select variants of particular interest were confirmed by directly examining the read pileup using the Java script tool Integrative Genomics Viewer (IGV) (64).

## Mutation definition

For the purpose of this work, we define a mutation as a single, discrete event leading to an observable change in genome sequence compared to the reference. As both coverage depth and read length vary significantly across our samples (see Table S1), we focus exclusively on mutations called with high confidence by breseq. For details on confidence thresholds and evidence types used to call mutations, please refer to the breseq manual: https://barricklab.org/twiki/pub/Lab/ToolsBacterialGenomeRese-quencing/documentation/methods.html. We excluded: (i) low-confidence calls, which can result from mixed non-clonal populations; (ii) unassigned missing coverage and unassigned junction evidence, which can result from large-scale deletions or chromo-somal rearrangements and/or highly repetitive sequences (32); (iii) certain high-quality calls with mutations in highly repetitive sequences. Mobile element mutations are identified solely as high-confidence mutations occurring inside IS elements and transposases (long-read sequencing data are required to resolve repeats and structural variation).

## ACKNOWLEDGMENTS

A.W.B.-F. is a Pew Scholar in the Biomedical Sciences, supported by The Pew Charitable Trusts. This work was also supported by grants from the National Science Foundation MCB 1936024 and 1651117 to A.K.S. The funders had no role in the study design, data collection and analysis, decision to publish, or preparation of the manuscript.

The authors thank Sierra Watson, Mar Martinez-Pastor, Cindy Darnell, and Angie Vreugdenil-Hayslette for preparing gDNA for sequencing many of the included strains. The authors thank Ron Peck for providing the H1209_RP strain for sequencing. The Marchfelder lab thanks the students of our master practical course 2014 and our bachelor and Erasmus students Nadja Raab, Jasmin Niederhofer, Stefan Werner, and Tereza Vavrdova for help with generating the deletion strains Δtrh7, Δtfb9, Δtfb12, ΔarcR14, and ΔarcR21. We thank the Duke Sequencing and Genomic Technologies core facility for their technical support with sequencing. We thank the Baliga lab for the transfer of *Hbt.* strains to the Schmid lab in 2009. We also thank the Schmid lab members for their support and critical feedback on the manuscript.

## AUTHOR AFFILIATIONS

[1]Department of Biology, Duke University, Durham, North Carolina, USA
[2]Computational Biology and Bioinformatics Graduate Program, Duke University, Durham, North Carolina, USA
[3]University Program in Genetics and Genomics, Duke University, Durham, North Carolina, USA
[4]Department of Microbiology and Cell Science, Institute of Food and Agricultural Sciences, University of Florida, Gainesville, Florida, USA
[5]Institut für Allgemeine Mikrobiologie, Christian-Albrechts-Universität Kiel, Kiel, Germany
[6]Department of Microbiology, The Ohio State University, Columbus, Ohio, USA
[7]Department of Biology, Rosenstiel Basic Medical Science Research Center, Brandeis University, Waltham, Massachusetts, USA
[8]Biology II, Ulm University, Ulm, Germany
[9]School of Life Sciences, Queen's Medical Centre, University of Nottingham, Nottingham, United Kingdom
[10]Center for Genomics and Computational Biology, Duke University, Durham, North Carolina, USA

## PRESENT ADDRESS

Rylee K. Hackley, Department of Microbiology, The Ohio State University, Columbus, Ohio, USA

Sungmin Hwang, Division of Convergence on Marine Science, National Korea Maritime and Ocean University, Busan, South Korea

Keely A. Dulmage, Precision Biosciences, Durham, North Carolina, USA

## AUTHOR ORCIDs

Peter Schönheit http://orcid.org/0000-0003-0915-8608
Anita Marchfelder http://orcid.org/0000-0002-1382-1794
Julie A. Maupin-Furlow http://orcid.org/0000-0001-6105-0923
Amy K. Schmid http://orcid.org/0000-0001-5821-8000

## FUNDING

| Funder | Grant(s) | Author(s) |
|---|---|---|
| National Science Foundation (NSF) | 1651117 | Amy K. Schmid |
| National Science Foundation (NSF) | 1936024 | Amy K. Schmid |
| Pew Charitable Trusts (PCT) | | Alexandre W. Bisson-Filho |

## AUTHOR CONTRIBUTIONS

Andrew L. Soborowski, Conceptualization, Data curation, Formal analysis, Investigation, Methodology, Software, Visualization, Writing – original draft | Rylee K. Hackley, Conceptualization, Data curation, Writing – review and editing | Sungmin Hwang, Data curation, Writing – review and editing | Guangyin Zhou, Data curation | Keely A. Dulmage, Data curation, Writing – review and editing | Peter Schönheit, Data curation, Resources, Writing – review and editing | Charles Daniels, Data curation, Resources, Writing – review and editing | Alexandre W. Bisson-Filho, Data curation, Resources, Writing – review and editing | Anita Marchfelder, Data curation, Resources, Writing – review and editing | Julie A. Maupin-Furlow, Data curation, Resources, Writing – review and editing | Thorsten Allers, Data curation, Resources, Writing – review and editing | Amy K. Schmid, Conceptualization, Data curation, Investigation, Methodology, Resources, Visualization, Writing – review and editing

## DATA AVAILABILITY

Novel sequencing data for this project were submitted to the National Center for Biotechnology Information (NCBI) Sequence Read Archive (SRA) and can be found under bioproject accession PRJNA1120443. SRA accessions for previously published sequencing data are available in Table S1. R code for performing analysis and generating figures is available at https://github.com/andrew-soborowski/halophile_genome_resequencing.

## ADDITIONAL FILES

The following material is available online.

### Supplemental Material

**Table S1 (mSystems01084-24-s0001.xlsx).** Strain, plasmid, and primer details.
**Table S2 (mSystems01084-24-s0002.xlsx).** Mutation information details for *Hbt. salinarum*.
**Table S3 (mSystems01084-24-s0003.xlsx).** Mutation information details for *Hfx. volcanii*.

## Open Peer Review

**PEER REVIEW HISTORY (review-history.pdf).** An accounting of the reviewer comments and feedback.

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
