## [Reviewer comments · mSystems]

Genomic re-sequencing reveals mutational divergence across genetically engineered strains of model archaea

Andrew Soborowski, Rylee Hackley, Sungmin Hwang, Guangyin Zhou, Keely Dulmage, Peter Schönheit, Charles Daniels, Alexandre Bisson Filho, Anita Marchfelder, Julie Maupin-Furlow, Thorsten Allers, and Amy Schmid

Corresponding Author(s): Amy Schmid, Duke University

Review Timeline:

Submission Date:	August 9, 2024
Editorial Decision:	August 29, 2024
Revision Received:	October 11, 2024
Accepted:	November 12, 2024

Editor: Pablo Ivan Nikel

Reviewer(s): Disclosure of reviewer identity is with reference to reviewer comments included in decision letter(s). The following individuals involved in review of your submission have agreed to reveal their identity: Kylie Allen (Reviewer #2); Houda Baati (Reviewer #3)

Transaction Report:

DOI: <https://doi.org/10.1128/msystems.01084-24>

Re: mSystems01084-24 (*Genomic re-sequencing reveals mutational divergence across genetically engineered strains of model archaea*)

Dear Prof. Schmid:
Dear Amy:

Thank you for the privilege of reviewing your work. Your submission to mSystems have been evaluated by three reviewers, who found the study to be of interest, and they suggested a numbers of modifications to improve the paper. You will find their comments at the bottom of this letter.

Revision Guidelines

Sincerely,

Prof. Pablo Ivan Nikel
Editor
mSystems

Reviewer #1 (Comments for the Author):

In their manuscript, entitled "Genomic re-sequencing reveals mutational divergence across genetically engineered strains of model archaea", Sobrowski et al. explore an important aspect of working with model organisms: genomic differences that can accumulate between strains from different laboratories and as unintended second-site mutations when generating genetically engineered strains. The effort taken by the involved groups is commendable, and the combination of re-sequencing and line

lineage reconstruction results in a valuable dataset that will aid future analyses. The emphasis on, and prevalence of, unintended second-site mutations in genetically engineered strains highlights the need for WGS of generated mutants. These results have the potential to establish new guidelines for re-sequencing newly generated mutant strains (or strains transferred between labs).

While there is little to criticize in this manuscript, the authors may want to consider the following minor comments and suggestions:

1. The interpretation that some mutations arose "during transfer" (l. 118, l. 218, l. 230) between labs requires a more detailed description of what is meant by that statement (only the physical transfer of the culture from one location to another, or does it include any culturing of the strain in either the lab of the sender or receiver?), how this conclusion was made (since sequencing was likely not performed right at the time of sending the strain nor at the time of receiving it), and how the strains were transferred (e.g. as glycerol stocks or on agar?).
2. From Figure 1, it is unclear how the timing of some mutations (position in the lineage diagram) was determined, since the figure suggests that not all strains were sequenced. E.g. an asterisk is placed between H119 and H53, as well as between H53 and H26_TA, but based on the dashed outlines, none of these strains were sequenced. From the text it becomes clear that H53 was sequenced previously, and that this information was taken into account, but the information about previously sequenced strains is missing in the figure. It also might be worth including in the figure the knockout strains that were sequenced here to visualize their lineages, and to clarify why some strains are included in the figure even though they were not sequenced. E.g. it is not immediately clear which strains are descendants of *ura3_AKS_2014* vs. *ura3_AKS_2017*, and that information is not included in Supplemental Table 1 either. For H26 from the Bisson lab, it seems unclear why it was included in the figure, since neither itself nor descendants of it were sequenced.
3. Details about how WGS information from previously sequenced strains was processed should be included, and it could be noted if a dedicated effort was made to aggregate WGS information from other labs and publications, or if only the lineages shown in Figure 1 were taken into account.
4. Supplemental Table 1: MDK407 is named as a background strain, but the genotype of MDK407 is not further defined.
5. Figure 3: the overlap of individual points turns some points practically invisible. E.g. 11 mutations are mentioned in the text for class I mutations in Hbt. but only 10 points are visible in the figure, likely due to an overlap of points.
6. Line 257: "Each strain averages 0.42 mutations." It should be clarified that this refers to second-site mutations.
7. Line 276: "However, Hfx. tends to accumulate more noncoding and synonymous mutations, accumulating 0.35 coding mutations per strain compared to only 0.32 in Hbt." This sentence should be revised, since it starts out with noncoding/synonymous mutations but then lists numbers for coding mutations that are similar (despite the "only" implying a marked difference).
8. Lines 368-371: Expanding on the recommendations to minimize or control for events that could lead to strain divergence (or unintended second-site mutations) could be beneficial to provide guidelines for the whole community. This could include more details on the sequencing and data analysis (see below).
9. Line 419: more details on the WGS should be provided. What was the procedure for adapter ligation and library preparation? Etc.
10. If possible, it would be helpful to provide the scripts for the bioinformatic data analysis from processing the raw data to identifying the mutations. This way, other labs could use this workflow as a template to follow the mentioned recommendations. The GitHub link provided in the Data Availability statement seems to only cover the generating of the figures.

Reviewer #2 (Comments for the Author):

Please see attached.

Reviewer #3 (Comments for the Author):

Some revisions are as follows: In the introduction section some references must be added related to the two archaeal genera studied: *Halobacterium salinarum*: <https://doi.org/10.1007/s00792-022-01273-0>; doi.org/10.3390/microorganisms11030587; <https://doi.org/10.1016/j.ygeno.2008.01.001>; DOI: 10.1002/mbo3.974. *Haloferax volcanii*: <https://doi.org/10.1128/JB.00690-20>; [doi: 10.3389/fmicb.2020.604926](https://doi.org/10.3389/fmicb.2020.604926); <https://doi.org/10.3390/genes12070963>.

In their manuscript, entitled “Genomic re-sequencing reveals mutational divergence across genetically engineered strains of model archaea”, Sobrowski et al. explore an important aspect of working with model organisms: genomic differences that can accumulate between strains from different laboratories and as unintended second-site mutations when generating genetically engineered strains. The effort taken by the involved groups is commendable, and the combination of re-sequencing and line lineage reconstruction results in a valuable dataset that will aid future analyses. The emphasis on, and prevalence of, unintended second-site mutations in genetically engineered strains highlights the need for WGS of generated mutants. These results have the potential to establish new guidelines for re-sequencing newly generated mutant strains (or strains transferred between labs).

While there is little to criticize in this manuscript, the authors may want to consider the following minor comments and suggestions:

1. The interpretation that some mutations arose “during transfer” (l. 118, l. 218, l. 230) between labs requires a more detailed description of what is meant by that statement (only the physical transfer of the culture from one location to another, or does it include any culturing of the strain in either the lab of the sender or receiver?), how this conclusion was made (since sequencing was likely not performed right at the time of sending the strain nor at the time of receiving it), and how the strains were transferred (e.g. as glycerol stocks or on agar?).
2. From Figure 1, it is unclear how the timing of some mutations (position in the lineage diagram) was determined, since the figure suggests that not all strains were sequenced. E.g. an asterisk is placed between H119 and H53, as well as between H53 and H26_TA, but based on the dashed outlines, none of these strains were sequenced. From the text it becomes clear that H53 was sequenced previously, and that this information was taken into account, but the information about previously sequenced strains is missing in the figure. It also might be worth including in the figure the knockout strains that were sequenced here to visualize their lineages, and to clarify why some strains are included in the figure even though they were not sequenced. E.g. it is not immediately clear which strains are descendants of *ura3_AKS_2014* vs. *ura3_AKS_2017*, and that information is not included in Supplemental Table 1 either. For H26 from the Bisson lab, it seems unclear why it was included in the figure, since neither itself nor descendants of it were sequenced.
3. Supplemental Table 1: MDK407 is named as a background strain, but the genotype of MDK407 is not further defined.
4. Figure 3: the overlap of individual points turns some points practically invisible. E.g. 11 mutations are mentioned in the text for class I mutations in Hbt. but only 10 points are visible in the figure, likely due to an overlap of points.
5. Line 257: “Each strain averages 0.42 mutations.” It should be clarified that this refers to second-site mutations.
6. Line 276: “However, Hfx. tends to accumulate more noncoding and synonymous mutations, accumulating 0.35 coding mutations per strain compared to only 0.32 in Hbt.” This sentence should be revised, since it starts out with noncoding/synonymous mutations but then lists numbers for coding mutations that are similar (despite the “only” implying a marked difference).
7. Lines 368-371: Expanding on the recommendations to minimize or control for events that could lead to strain divergence (or unintended second-site mutations) could be

beneficial to provide guidelines for the whole community. This could include more details on the sequencing and data analysis (see below).

8. Line 419: more details on the WGS should be provided. What was the procedure for adapter ligation and library preparation? Etc.
9. If possible, it would be helpful to provide the scripts for the bioinformatic data analysis from processing the raw data to identifying the mutations. This way, other labs could use this workflow as a template to follow the mentioned recommendations. The GitHub link provided in the Data Availability statement seems to only cover the generating of the figures.

Review response, mSystems01084-24: Genomic re-sequencing reveals mutational divergence across genetically engineered strains of model archaea

We appreciate the thorough and helpful review of our study. We have made minor revisions and clarifications to the figures and text in response to reviewer suggestions. We find that these revisions aided the clarity of the manuscript. The conclusions of the study remain unchanged. Below we respond in detail to reviewer concerns point by point, with reviewer comments marked in italicized font and our responses in plain font.

Reviewer #1 (Comments for the Author):

In their manuscript, entitled "Genomic re-sequencing reveals mutational divergence across genetically engineered strains of model archaea", Sobrowski et al. explore an important aspect of working with model organisms: genomic differences that can accumulate between strains from different laboratories and as unintended second-site mutations when generating genetically engineered strains. The effort taken by the involved groups is commendable, and the combination of re-sequencing and line lineage reconstruction results in a valuable dataset that will aid future analyses. The emphasis on, and prevalence of, unintended second-site mutations in genetically engineered strains highlights the need for WGS of generated mutants. These results have the potential to establish new guidelines for re-sequencing newly generated mutant strains (or strains transferred between labs).

We appreciate the positive feedback and are grateful that the resource is seen as useful to the community.

While there is little to criticize in this manuscript, the authors may want to consider the following minor comments and suggestions:

1. The interpretation that some mutations arose "during transfer" (l. 118, l. 218, l. 230) between labs requires a more detailed description of what is meant by that statement (only the physical transfer of the culture from one location to another, or does it include any culturing of the strain in either the lab of the sender or receiver?), how this conclusion was made (since sequencing was likely not performed right at the time of sending the strain nor at the time of receiving it), and how the strains were transferred (e.g. as glycerol stocks or on agar?).

Detailed information was added in the introduction to describe the procedures typically used to transfer strains between labs (lines 60-65). Additionally, at the first mention of a mutation arising during transfer, we expanded upon the procedures of transfer in more detail in the results.

*2. From Figure 1, it is unclear how the timing of some mutations (position in the lineage diagram) was determined, since the figure suggests that not all strains were sequenced. E.g. an asterisk is placed between H119 and H53, as well as between H53 and H26_TA, but based on the dashed outlines, none of these strains were sequenced. From the text it becomes clear that H53 was sequenced previously, and that this information was taken into account, but the information about previously sequenced strains is missing in the figure. It also might be worth including in the figure the knockout strains that were sequenced here to visualize their lineages, and to clarify why some strains are included in the figure even though they were not sequenced. E.g. it is not immediately clear which strains are descendants of *ura3_AKS_2014* vs. *ura3_AKS_2017*, and that information is not included in Supplemental Table 1 either. For H26*

from the Bisson lab, it seems unclear why it was included in the figure, since neither itself nor descendants of it were sequenced.

Thank you for this feedback on the clarity of the lineage tracing section and figure 1 as a whole. Regarding the timing of mutations, it is correct that not every strain was sequenced, and it is thus not straightforward to assign mutations to specific points in time. Given sequenced outgroup strains, such as the Daniels lab $\Delta pyrF$ strain or the Bisson lab DS2 strain, we were able to localize mutations in time between certain events. For example, for the two mutations listed in the Dyall Smith Lab, they are present in the Daniels lab $\Delta pyrF$ strain, and thus must have preceded that branch point, but they are absent in the Bisson lab DS2 strain, and thus developed after the two DS2 lineages split. Especially for the older mutations, we don't have sufficient information on which precise event resulted in the observed mutation. For example, for the four mutations unique to the Daniels lab $\Delta pyrF$ strain, given incomplete lineage tracing information and only the endpoint strain sequenced, we can only place the mutations as having occurred after the DS70 split in the Dyall Smith lab. In these cases, we simply group the asterisks at the earliest point they could have occurred. In the case of the Allers lab strains, we do have a mutational analysis for H26_TA and H53 that allowed us to localize two mutations more accurately. This has been clarified in the figure 1 legend: "Asterisks denote individual mutations and are placed at earliest transition they may have occurred" (pg. 5).

We decided to exclude knockout strains from this figure to preserve clarity: the number of knockouts significantly dwarfs the number of parental strains depicted here. We have updated the labelling in supplementary table 1 to be consistent with the labels we introduce in the manuscript as well as the labels used widely in the field. In our view, these revisions should allow easier correspondence between the parental strains on this figure and the analyzed knockouts.

3. Details about how WGS information from previously sequenced strains was processed should be included, and it could be noted if a dedicated effort was made to aggregate WGS information from other labs and publications, or if only the lineages shown in Figure 1 were taken into account.

To address this potential confusion, we have clarified in the methods section that WGS information from previously sequenced strains was processed using the same procedures as data obtained for this study. For *Haloferax*, some WGS information was also incorporated from the Allers lab parental strains (H53_TA and H119 from Hawkins et al. 2014); however, our searches failed to find publicly available (for example, on NCBI Sequence Read Archive) and usable (i.e. of similar read depth, sequencing platform, and quality) pre-existing WGS datasets outside of the Schmid Lab. In the case of *Halobacterium*, due to the relatively small number of knockout lineages available to us, we restricted our analysis to the Baliga lab NRC-1 lineage.

4. Supplemental Table 1: MDK407 is named as a background strain, but the genotype of MDK407 is not further defined.

Thank you for noting this inconsistency. In the revised Supplemental Table 1, we have moved the designation for MDK407 (note the typo correction to MPK407 as per the original publication of the strain in Peck et al., 2000) to that of the published strain name for $\Delta ura3$. Under background strains, we have replaced MPK407 with *ura3_AKS_2014*, *ura3_AKS_2017*, or *ura3_NB* as appropriate to identify the specific $\Delta ura3$ knockout strain used to create each of the downstream knockouts.

5. *Figure 3: the overlap of individual points turns some points practically invisible. E.g. 11 mutations are mentioned in the text for class I mutations in Hbt. but only 10 points are visible in the figure, likely due to an overlap of points.*

We have manually adjusted the jitter we imposed on certain points in the y-axis (which is otherwise meaningless outside of visual clarity) for this figure. Most points are now visibly distinct. In the case of a pileup at a short genomic range, we have stratified the points such that all are now discernable.

6. *Line 257: "Each strain averages 0.42 mutations." It should be clarified that this refers to second-site mutations.*

Thank you for pointing out this potential confusion, we have clarified it in the text by saying each strain averaged 0.42 unexpected second site mutations.

7. *Line 276: "However, Hfx. tends to accumulate more noncoding and synonymous mutations, accumulating 0.35 coding mutations per strain compared to only 0.32 in Hbt." This sentence should be revised, since it starts out with noncoding/synonymous mutations but then lists numbers for coding mutations that are similar (despite the "only" implying a marked difference).*

To address this, we have revised the paragraph to highlight the difference in synonymous/noncoding mutational density between species, while coding mutation density is similar (lines 282-298 in the revised manuscript).

8. *Lines 368-371: Expanding on the recommendations to minimize or control for events that could lead to strain divergence (or unintended second-site mutations) could be beneficial to provide guidelines for the whole community. This could include more details on the sequencing and data analysis (see below).*

This feedback on the concluding paragraph was helpful. In the revised final paragraph of the discussion, we have clarified and briefly expanded upon some measures that labs can easily take to minimize and control divergence events and emphasized how this will be of benefit to the scientific community as a whole (lines 384-396 of revised manuscript).

9. *Line 419: more details on the WGS should be provided. What was the procedure for adapter ligation and library preparation? Etc.*

Thank you for pointing out this omission, we have added details on the kit and adapters used to the methods section (line 453-458 of revised manuscript).

10. *If possible, it would be helpful to provide the scripts for the bioinformatic data analysis from processing the raw data to identifying the mutations. This way, other labs could use this workflow as a template to follow the mentioned recommendations. The GitHub link provided in the Data Availability statement seems to only cover the generating of the figures.*

Thank you for this request. We have provided a sample bash script file at the same github link walking through the key commands in the analysis. Additionally, we have provided .yaml files on github with full conda environments that were used to perform this analysis. A caution in using these scripts, however, is that the exact scripts used will differ for different users, as they depend heavily on computing platform, cluster setup, and specific species/file names. We have noted this caution in the readme portion of the github repository.

Reviewer #2 (Comments for the Author) and response:

The manuscript by Soborowski et al. reports the results and analysis of whole-genome sequencing data from 60 strains of two species of Haloarchaea from several different labs. This provided key insights into mutations occurring during transfers between labs and during experimenter-intended genetic manipulations. This information is highly impactful for the Haloarchaeal research community, but also for the microbial research community at large. The potential accumulation of mutations in various lab strains can be a major problem with reproducibility between research groups as well as with proper interpretation of phenotypes etc. in strains with intended mutations. This is something that is not discussed enough. The rigorous analysis reported in this manuscript will set the stage for other microbial research groups to strongly consider WGS as a key tool for maintaining the integrity of results from experiments with model microbes. Overall, the manuscript is very well-written and well-presented.

We appreciate the reviewer's positive feedback.

I have a few comments for the authors to consider:

1. First sentence of abstract and first sentence of importance: "Archaea are important due to their shared evolutionary history with eukaryotes". This is quite a narrow view of why archaea are important. For example, methanogens and anaerobic methano(alkano)trophs play key roles in the global carbon cycle and in regulating our climate. Thus, the first sentence of the importance section especially should be modified.

Thank you for this important feedback on the opening sentences. We have modified the abstract and importance section to highlight the diverse role that archaea play, while also placing more emphasis on our focus on halophiles as a model for archaea

2. Figure 2a: It looks like the heat map may have an error for the $\Delta hpyA\Delta mc1$ and $\Delta mc1$ strains. The blue is the "experimenter-intended mutation", which would mean the double knockout should have two blue squares while the single knockout should have one. Maybe the labels are just switched. Please check this.

We thank the reviewer for catching this mistake. We have verified that these labels were switched and have corrected them in the updated Figure 2a. We have also verified the correctness of all other labels in this figure.

*3. For discussion: Assuming I am interpreting figure 2 and figure 3 correctly, it looks like several of the unintended class III mutations (especially in *H. volcanii*) are located near the experimenter-intended mutation (knockout) on the genome. It would be nice for the authors to include a discussion on the relevance of this and whether other genome-editing methods (i.e. CRISPR/Cas) should/could be developed for Haloarchaea to minimize this.*

This is an interesting observation. We looked back the raw data for mutation calls and found that, despite the appearance of mutations stacking up near intended knockout locations, 50,000 bp was the closest distance between any given unintended mutation in a knockout (KO) strain in *Hfx.* and the knockout site of interest for that specific strain. Only found 2 such cases were found. These include an intergenic single base deletion in the $\Delta pai2$ KO and a coding SNP in the HVO_2423 knockout. Given these results, we decided that a discussion of alternative genome-editing methods were not pertinent at this time. Instead, we've added a clarification in the results section that unexpected mutations do not tend to accumulate at knockout sites (lines 288-289).

4. For discussion: It is briefly mentioned in the introduction, but it would be helpful to include a more comprehensive discussion where the results for Haloarchaea mutations accumulating in lab strains are compared to model bacteria and model eukaryotes if these data are available.

Are the archaea accumulating mutations less or more compared to bacteria? If so, is this a result of their biology and/or more likely due to differences among lab procedures? Also, the intro states that “this problem has yet to be addressed in haloarchaeal species”. Has this problem been discussed or experimentally addressed for any other archaeal species (i.e. Sulfolobus or methanogens)?

This is a really interesting point that is worth investigating in future research. Unfortunately, to the best of our knowledge, at this point these results are generally not available for any other model system, archaea included. From our research, some of the most thorough analysis has been done tracing the lineage of *B. subtilis* (see Zeigler et al. 2008), though the issue of divergent lab strains is at least recognized or worked around in other common lab systems such as *E. coli* (see in-text citations 30 and 31 in the introduction). At a surface level, it appears that halophiles have accumulated fewer mutations in their parental strains compared to *B. subtilis*, though both methodology and biology differ greatly between the species. *B. subtilis* has been circulated as a model organism since the 1940s among a very large number of labs, compared to the more recent development of haloarchaea as model organisms and considerably reduced circulation. It is not clear from the analysis that has been done how differences in lab practice among *B. subtilis* researchers may have affected mutation accumulation within the species, making it difficult to say how lab practice versus innate biology may affect accumulation between species. It is also not clear how many intermediate steps each studied *B. subtilis* strain went through between its ancestral and modern stocks, making direct comparison of lab stock mutation rates impossible given current data. A comprehensive re-sequencing study comparing mutation accumulation rates would be necessary and therefore out of scope of the current manuscript. We therefore chose not to comment on this in the discussion.

5. For Figure 3, recommend using different colors to classify the different types of mutations. For example, the green is used to denote class III mutations in figure 2, but then used to denote non-synonymous mutations in figure 3. This got a bit confusing when going back and forth between the figures.

Thank you for this comment on the clarity of figure 3. We have adjusted the colors used in this figure to be more easily differentiable from those of figure 2.

Reviewer #3 (Comments for the Author) and response:

Some revisions are as follows: In the introduction section some references must be added related to the two archaeal genera studied: Halobacterium salinarum: <https://doi.org/10.1007/s00792-022-01273-0>; doi.org/10.3390/microorganisms11030587; <https://doi.org/10.1016/j.ygeno.2008.01.001>; DOI: 10.1002/mbo3.974. Haloferax volcanii: <https://doi.org/10.1128/JB.00690-20>; doi: 10.3389/fmicb.2020.604926; <https://doi.org/10.3390/genes12070963>.

Thank you for suggesting these papers highlighting the rich study of halophiles and the comparisons between strains NRC-1 and R1. We have cited these papers to highlight the various interests the community has in halophile study in the introduction (line 4). We also cite the two suggested Pfeiffer et al papers (2008, 2020, lines 12-14) to clarify that our study focuses on strain NRC-1, while acknowledging that genomic comparisons between NRC-1 and R1 strains have been performed previously.

Re: mSystems01084-24R1 (*Genomic re-sequencing reveals mutational divergence across genetically engineered strains of model archaea*)

Dear Prof. Amy K Schmid:

I am delighted to report that your manuscript has been accepted, and I am forwarding it to the ASM production staff for publication. Your paper will first be checked to make sure all elements meet the technical requirements. ASM staff will contact you if anything needs to be revised before copyediting and production can begin. Otherwise, you will be notified when your proofs are ready to be viewed.

Cover Image Submissions: If you would like to submit a potential Cover Image, please email a file and a short legend to mSystems@asmusa.org. Please note that we can only consider images that (i) the authors created or own and (ii) have not been previously published. By submitting, you agree that the image can be used under the same terms as the published article. Image File requirements: TIF/EPS, 7.5 inches wide by 8.25 inches tall (at least 2,250 pixels wide by 2,475 pixels tall), minimum 300 dpi resolution (600 dpi preferred), RGB, and no figure elements, e.g., arrows or panel labels. The legend should be a short description of the image, 1-2 sentences recommended. Please download and use this interactive template in Adobe to ensure that your proposed cover image meets our size requirements (<https://journals.asm.org/pb-assets/pdf-text-excel-files/ASM-Interactive-Sizing-Cover-Template-1715689791.pdf>).

Sincerely,

Prof. Pablo Ivan Nickel
Editor
mSystems

I write you in regards to manuscript (mSystems01084-24)"Genomic re-sequencing reveals mutational divergence across genetically engineered strains of model archaea" which was submitted to mSystems.

The manuscript is suitable for publication. All comments and questions have been taken into consideration.

Sincerely,

Dr Houda Baati